# Reinforcement Learning
# with Non-Exponential Discounting

**Matthias Schultheis**
Centre for Cognitive Science
Technische Universität Darmstadt
matthias.schultheis@tu-darmstadt.de

**Constantin A. Rothkopf**
Centre for Cognitive Science
Technische Universität Darmstadt
constantin.rothkopf@tu-darmstadt.de

**Heinz Koeppl**
Centre for Cognitive Science
Technische Universität Darmstadt
heinz.koeppl@tu-darmstadt.de

## Abstract

Commonly in reinforcement learning (RL), rewards are discounted over time using an exponential function to model time preference, thereby bounding the expected long-term reward. In contrast, in economics and psychology, it has been shown that humans often adopt a hyperbolic discounting scheme, which is optimal when a specific task termination time distribution is assumed. In this work, we propose a theory for continuous-time model-based reinforcement learning generalized to arbitrary discount functions. This formulation covers the case in which there is a non-exponential random termination time. We derive a Hamilton–Jacobi–Bellman (HJB) equation characterizing the optimal policy and describe how it can be solved using a collocation method, which uses deep learning for function approximation. Further, we show how the inverse RL problem can be approached, in which one tries to recover properties of the discount function given decision data. We validate the applicability of our proposed approach on two simulated problems. Our approach opens the way for the analysis of human discounting in sequential decision-making tasks.

## 1 Introduction

An often observed phenomenon in humans and animals is that they prefer rewards rather sooner than later [1]. It comes with no surprise that an animal searching for food aims to find it as soon as possible and employees working in a company prefer to be paid after each month instead of after each year. In behavioral experiments, it was observed that people are even willing to pay a price to receive rewards earlier [2]. It can be concluded that rewards become of less value in the future, a phenomenon which is known as discounting [2].

Modeling and experimentally inferring discounting functions, which describe how values are discounted over time, already have a long history in economics and psychology, where many different functional forms have been proposed [3, 4, 5]. While from an economic perspective, a fixed interest rate seems reasonable, leading to an exponential discount function, human behavior is oftentimes better described by a hyperbolic curve [6]. The reason for this is that human decisions are typically not consistent regarding shifting rewards in time, a phenomenon named preference reversal: We might find that a subject prefers a smaller reward on the same day over a larger reward one day after. However, when given the choice between the same smaller reward in 365 days, compared to the same

36th Conference on Neural Information Processing Systems (NeurIPS 2022).

larger reward in 366 days, the subject is more likely to be willing to wait one day more for the larger reward.

While some literature has branded the observed discounting behavior as not being rational [7], there has been an increasing amount of work identifying circumstances under which this behavior is indeed optimal [8, 9, 10]. One widely-adopted theory rationalizing hyperbolic discounting is to assume a constant risk for the reward to become unavailable but with the risk being uncertain [11]. Under this condition, one should adapt the preference over time, as with time the expected risk decreases.

Discounting is also widely applied in the field of reinforcement learning (RL) and optimal control [12, 13, 14]. First, when modeling infinite horizon time objectives, a discount function is needed to make the expected long-term reward objective well-defined, as otherwise, it would become infinite. Second, for autonomous agents, it also makes sense to model a preference for earlier rewards in order to find shortest paths to save time and energy. Third, the discount function can be interpreted as the probability of termination inducing a specific end-time distribution [15, 12].

Despite these obvious connections, discounting models of psychology and reinforcement learning have remained mostly independent with few exceptions [16, 17]. Generalizing reinforcement learning to a broader range of discount functions would enable solutions for applications in which the end time follows a specific distribution. On the other hand, methods for determining optimal decisions in sequential decision-making tasks with general end-time distributions would provide tools which can help to explain human decision-making under uncertainty.

In this work, we present a theory for model-based reinforcement learning in continuous time based on non-exponential discount functions. First, we investigate the conditions under which the objective of maximizing the long-term reward formulated with hyperbolic discounting is well-defined. Second, we derive a Hamilton–Jacobi–Bellman (HJB)-type equation for a general discount function and describe how to solve it to obtain the optimal policy. Third, we provide an approach to tackle the inverse reinforcement learning (IRL) problem, to estimate parameters of the discount function given decision data. Finally, we show the applicability of our proposed method on two simulated problems.

## 2 Related work

Optimal control in continuous time and space has a long history with many classical works [18, 19, 20, 21]. Continuous-time reinforcement learning formulations have been developed [14] and various solution methods have been proposed [22, 23]. Solution approaches that solve the HJB equation directly include linearization techniques [24, 25], path integral formulations [26, 27, 12], and collocation-based methods [28]. In recent years, it has become increasingly popular to use neural networks for function approximation to solve the HJB equation [29, 30, 31, 32, 33].

Non-exponential discounting was considered in literature in behavioral economics, psychology, and neuroscience [4, 34, 2]. For humans and animals, preference reversal behavior has been described [35, 36, 5] and different functional forms for the discount function have been proposed [3, 37]. In most works, only non-sequential decision-making tasks have been studied, i.e., situations with a single decision for each independent trial. In this line of work, the method of limits has been used to elicit discount functions based on binary decisions [38, 39, 40] or an adjustment method has been applied [41]. Some methods have been proposed that specifically aim to efficiently estimate the discount function for binary choice responses [42, 43]. There have been a few exceptions for which decision trials were actually dependent but could still be considered independently for the analysis [44, 45]. Further, there has been research aiming to find rational explanations for the encountered discounting behavior [8, 9, 10], e.g., by assuming uncertainty about a constant hazard [11].

Decision processes with non-exponential time distributions have been studied in the field of semi-Markov Decision Processes (SMDPs) [46, 47, 48, 49], where transition times between discrete states can follow arbitrary distributions. MDPs with quasi-hyperbolic discounting, for which all future rewards are additionally discounted by a constant factor, have been addressed in [50, 51, 52]. They can be used to model preference reversal but are limited to the specific form assumed. Stochastic processes with quasi-hyperbolic discounting have been considered in financial economics for portfolio management [53, 54, 55, 56].

More closely related to our work, Fedus et al. [17] presented an approximate method for solving MDPs with hyperbolic discounting. They approached the problem by solving the corresponding

exponentially-discounted problem for many different discount factors and combining the results. Part of the approximation, however, is that the value function and policy are static, failing to model preference reversals over time. Another line of work considers MDPs with discount factors that are coupled to the value function to imitate hyperbolic discounting behavior [16, 57]. As time is only considered indirectly through the magnitude of the value function, these approaches cannot be used for finding the solution to a given specific discount function or eliciting the discount function from data.

Finally, inverse reinforcement learning approaches have been mainly used to learn reward functions given data [58, 59, 60, 61]. These methods were also applied to learn properties of human behavior [62, 63, 64, 65]. Other inverse approaches have focused on learning dynamics models [66] or learning rules [67] from sequential decision-making data.

## 3 Background

### 3.1 Survival analysis

In survival analysis [68], one is interested in the duration until events occur. In its classical form one considers a single event, for which the duration can be described as a continuous random variable $T$ with cumulative distribution function $F(t) = P(T \leq t)$ and probability density function $f(t)$, where $t \in \mathbb{R}_0^+$ denotes the elapsed time. In survival analysis literature, $F(t)$ is known as the failure function and the survival function is defined as $S(t) = 1 - F(t) = P(T > t)$. The survival function is monotonically decreasing and has the properties $S(0) = 1$ and $\lim_{t \to \infty} S(t) = 0$. By the conditioning rule, one finds $P(T > t_1 \mid T > t_0) = S(t_1)/S(t_0)$ if $t_1 > t_0$. The hazard rate is defined as $\alpha(t) = \lim_{\Delta t \to 0} \frac{1}{\Delta t} P(t \leq T < t + \Delta t \mid t \leq T)$, yielding the relations

$$\alpha(t) = \lim_{\Delta t \to 0} \frac{1}{\Delta t} \frac{S(t) - S(t + \Delta t)}{S(t)} = -\frac{S'(t)}{S(t)} \quad \text{and} \quad S(t) = \exp\left(-\int_0^t \alpha(\tau)\, \mathrm{d}\tau\right). \quad (1)$$

For a constant hazard rate $\alpha(t) = \lambda$, one finds $S(t) = \exp(-\lambda t)$, which can be shown to be the unique memory-less survival function [68], i.e., $P(T > t + \Delta t \mid T > t) = P(T > \Delta t)$ for all $t, \Delta t \in \mathbb{R}_0^+$.

### 3.2 Discounting and preference reversal

We consider the setting in which a subject collects a single reward and shows a form of time preference, i.e., rewards are desired rather sooner than later. We model the value of a reward $r \in \mathbb{R}$ as a function $L : \mathbb{R} \times \mathbb{R}_0^+ \to \mathbb{R}$ with $L(r, t) = S(t) \cdot r$, where $S(t)$ is the discount function decreasing with time $t \in \mathbb{R}_0^+$. With the convention that $S(t_0) = 1$ and $\lim_{t \to \infty} S(t) = 0$, we can regard $S(t)$ as the survival function (Section 3.1), with the interpretation that the reward becomes unavailable with hazard rate $\alpha(t)$ [11].

When assuming a constant hazard rate $\alpha(t) = \lambda$, we say that the subject discounts exponentially, as $S(t) = \exp(-\lambda t)$. By the memory-less property, we have that if the subject prefers reward $r_1$ after $t_1$ over $r_2$ after $t_2$, she or he would remain consistent with the election if we presented the choice again later in time. On the other hand, if we assume a constant but unknown hazard rate $\lambda$ with belief $p(\lambda) = \mathrm{Gamma}(\lambda; \alpha_0, \beta_0)$, we obtain a hyperbolic form for the expected survival function:

$$S(t; \alpha_0, \beta_0) = \int_\lambda \exp(-\lambda t)\, p(\lambda)\, \mathrm{d}\lambda = \frac{1}{(\frac{t}{\beta_0} + 1)^{\alpha_0}} \quad (2)$$

The posterior belief over $\lambda$ at a later point in time can be derived using Bayes rule and is given by $p(\lambda \mid t) = \mathrm{Gamma}(\lambda; \alpha_0, \beta_0 + t)$. The expected hazard rate $\alpha(t)$ is given by the posterior mean,

$$\alpha(t) = \int_\lambda \lambda\, p(\lambda \mid t)\, \mathrm{d}\lambda = \frac{\alpha_0}{\beta_0 + t}. \quad (3)$$

For discount functions other than the exponential, such as the hyperbolic discount function in Eq. (2), the hazard rate varies over time and preferences among options may change.

### 3.3 Optimal control

In stochastic optimal control [69], we consider a Markovian system with continuous state $\mathbf{x}(t) \in \mathbb{R}^n$ evolving according to the stochastic differential equation (SDE) $\mathrm{d}\mathbf{X}(t) = f(\mathbf{X}(t), \mathbf{u}(t)) \, \mathrm{d}t + G(\mathbf{X}(t), \mathbf{u}(t)) \, \mathrm{d}\mathbf{W}(t)$ where $t \in \mathbb{R}_0^+$ denotes time, $f : \mathcal{X} \times \mathcal{U} \to \mathcal{X}$ the drift function, $G : \mathcal{X} \times \mathcal{U} \to \mathcal{X} \times \mathbb{R}^m$ the dispersion matrix, and $\mathbf{W}(t) \in \mathbb{R}^m$ $m$-dimensional Brownian motion. The goal of optimal control is to determine the control inputs $\mathbf{u}(t) \in \mathcal{U}$ given the current state $\mathbf{x}$ at time $t$, in order to maximize the expected long term discounted reward with reward function $R : \mathcal{X} \times \mathcal{U} \to \mathbb{R}$. The solution is characterized by the optimal value function,

$$V^*(\mathbf{x}) = \max_{\mathbf{u}_{[t,\infty)}} \mathbb{E}\left[ \int_t^\infty \exp(-\lambda(\tau - t)) \, R(\mathbf{X}(\tau), \mathbf{u}(\tau)) \, \mathrm{d}\tau \,\Big|\, \mathbf{X}(t) = \mathbf{x} \right], \tag{4}$$

where maximization is carried out over all trajectories $\mathbf{u}_{[t,\infty)} := \{\mathbf{u}(\tau)\}_{\tau \in [t,\infty]}$. The quantity $\exp(-\lambda t)$ is the discount factor, which ensures convergence of the integral and models a preference for earlier rewards. $\lambda$ can be interpreted in terms of survival analysis as the hazard rate for termination (cf. Section 3.1). According to the principle of optimality, the stochastic Hamilton-Jacobi-Bellman (HJB) equation, given by

$$\lambda V^*(\mathbf{x}) = \max_{\mathbf{u} \in \mathcal{U}} \left[ R(\mathbf{x}, u) + V_\mathbf{x}^*(\mathbf{x})^T f(\mathbf{x}, \mathbf{u}) + \frac{1}{2} \operatorname{tr} \left\{ V_{\mathbf{xx}}^*(\mathbf{x}) G(\mathbf{x}, \mathbf{u}) G(\mathbf{x}, \mathbf{u})^T \right\} \right],$$

provides a condition for the optimal value function. Here, partial derivatives are denoted by the index notation, i.e., $V_\mathbf{x}$ denotes the partial derivative of $V$ w.r.t. $\mathbf{x}$ and $V_{\mathbf{xx}}$ the respective Hessian. Note that the optimal value function depends on time only indirectly through the state $\mathbf{x}$ due to the Markov property and the memory-less discount function. This dependence also applies to the optimal policy $\pi^* : \mathcal{X} \to \mathcal{U}$, given by the maximizer of the right-hand side of the HJB equation, i.e.,

$$\pi(\mathbf{x}) = \arg \max_{\mathbf{u} \in \mathcal{U}} \left[ R(\mathbf{x}, u) + V_\mathbf{x}^*(\mathbf{x})^T f(\mathbf{x}, \mathbf{u}) + \frac{1}{2} V_{\mathbf{xx}}^*(\mathbf{x}) G(\mathbf{x}, \mathbf{u}) G(\mathbf{x}, \mathbf{u})^T \right].$$

## 4 Reinforcement learning with general discount function

We consider a system as in stochastic optimal control (cf. Section 3.3) with continuous state space $\mathbb{R}^n$ and finite set of controls $\mathcal{U}$. Instead of an exponential discount function, we allow for general survival functions $S(t)$ based on a time-dependent hazard rate $\alpha(t)$. This setting generalizes the discounting formulation of Section 3.2 to sequential decisions. As we will see later, the resulting value function will become time-dependent in contrast to Section 3.3 and we can further allow for time-dependent dynamics and reward without increasing complexity. We therefore assume for the state evolution the SDE $\mathrm{d}\mathbf{X}(t) = f(\mathbf{X}(t), \mathbf{u}(t), t) \, \mathrm{d}t + G(\mathbf{X}(t), \mathbf{u}(t), t) \, \mathrm{d}\mathbf{W}(t)$ with $f : \mathcal{X} \times \mathcal{U} \times \mathbb{R}_0^+ \to \mathcal{X}$ and $G : \mathcal{X} \times \mathcal{U} \times \mathbb{R}_0^+ \to \mathcal{X} \times \mathbb{R}^m$. The objective function measuring the total expected discounted reward is given by

$$J\left(\mathbf{u}_{[0,\infty)}\right) = \mathbb{E}\left[ \int_0^\infty S(\tau) \, R(\mathbf{X}(\tau), \mathbf{u}(\tau), \tau) \, \mathrm{d}\tau \right], \tag{5}$$

with time-dependent reward function $R : \mathcal{X} \times \mathcal{U} \times \mathbb{R}_0^+ \to \mathbb{R}$. Analogous to Section 3.3, we define the expected reward-to-go as the value function

$$V^*(\mathbf{x}, t) = \max_{\mathbf{u}_{[t,\infty)}} \mathbb{E}\left[ \int_t^\infty \frac{S(\tau)}{S(t)} R(\mathbf{X}(\tau), \mathbf{u}(\tau), \tau) \, \mathrm{d}\tau \,\Big|\, \mathbf{X}(t) = \mathbf{x} \right], \tag{6}$$

where $S(\tau)/S(t)$ is the probability of survival until time $\tau$, conditioned on the fact that one already has survived until time $t$ (cf. Section 3.1). In contrast to Eq. (4), the value function becomes time-dependent through the general survival function and also the optimal policy depends on time.

### 4.1 Technical requirements of model and discount function

First, the stochastic process defined in Section 4 must have a strongly unique solution, which is the case if $f$ and $G$ grow at most linearly in $\mathbf{X}$ and are Lipschitz continuous in the same variable [70]. Further, the optimal value function is only in rare cases smooth enough to be a solution in the

"classical" sense. Instead, one considers a viscosity solution [20], which satisfies the HJB equation in an appropriate generalized sense. A sufficient condition for the existence of such a solution is that $f$ and $G$ are continuous with bounded continuous first derivatives w.r.t. $\mathbf{X}$ and $t$, and bounded. $R$ and $S$ need to be continuous and grow at most polynomially in $\mathbf{X}$ and $t$ in an absolute sense. More details on the existence and uniqueness of viscosity solutions can be found in [20]. A third requirement is that the integral in the definition of the value function Eq. (5) converges. For the case of a hyperbolic discount function as in Eq. (2), we find the following theorem:

**Theorem 1.** *Consider the hyperbolic discount function in Eq. (2). If the reward function $R(\mathbf{x}, \mathbf{u}, t)$ is bounded above for all $(\mathbf{x}, \mathbf{u}, t) \in \mathcal{X} \times \mathcal{U} \times \mathbb{R}_0^+$, and $\alpha_0 > 1$, the value function defined in equation Eq. (6) is well-defined. If $R(\mathbf{x}, \mathbf{u}, t)$ is bounded below for all $(\mathbf{x}, \mathbf{u}, t) \in \mathcal{X} \times \mathcal{U} \times \mathbb{R}_0^+$, and $\alpha_0 \leq 1$, the value function is not well-defined as it becomes infinite.*

*Proof.* See Appendix A.

In the examples considered later, we will assume a bounded reward function and a hyperbolic discount function with $\alpha_0 > 1$, for which Eq. (5) and Eq. (6) are well-defined.

## 4.2 HJB equation for a general discount function

In the following, we give a brief overview of the derivation of the HJB equation for a general discount function. A more detailed derivation is provided in Appendix B. First, we split the integral in Eq. (6) into two terms such that we obtain a recursive formulation of the value function:

$$
V^*(\mathbf{x}, t) = \max_{\mathbf{u}_{[t, t+\Delta t]}} \mathbb{E} \left[ \int_t^{t+\Delta t} \frac{S(\tau)}{S(t)} R(\mathbf{X}(\tau), \mathbf{u}(\tau), \tau) \, \mathrm{d}\tau \right.
$$

$$
\left. + \frac{S(t + \Delta t)}{S(t)} V^*(\mathbf{X}(t + \Delta t), t + \Delta t) \, \Big| \, \mathbf{X}(t) = \mathbf{x} \right]
$$

For the second term in the expectation, we apply a Taylor expansion and Itô's formula [69] and obtain

$$
V^*(\mathbf{X}(t + \Delta t), t + \Delta t) = V^*(\mathbf{X}(t), t) + \int_t^{t+\Delta t} V_{\mathbf{x}}^*(\mathbf{X}(\tau), \tau) \, f(\mathbf{X}(\tau), \mathbf{u}(\tau), \tau) \, \mathrm{d}\tau
$$

$$
+ \int_t^{t+\Delta t} V_{\mathbf{x}}^*(\mathbf{X}(\tau), \tau) \, G(\mathbf{X}(\tau), \mathbf{u}(\tau), \tau) \, \mathrm{d}\mathbf{W}(\tau) + \int_t^{t+\Delta t} V_t^*(\mathbf{X}(\tau), \tau) \, \mathrm{d}\tau
$$

$$
+ \int_t^{t+\Delta t} \frac{1}{2} \operatorname{tr} \left\{ V_{\mathbf{xx}}^*(\mathbf{X}(\tau), \tau) \, G(\mathbf{X}(\tau), \mathbf{u}(\tau), \tau) \, G(\mathbf{X}(\tau), \mathbf{u}(\tau), \tau)^T \right\} \mathrm{d}\tau + o(\Delta t).
$$

Plugging this result into the equation above, dividing both sides by $\Delta t$, and taking the limit $\Delta t \to 0$, as well as calculating the expectation w.r.t. $\mathbf{W}(t)$ leads to the desired HJB equation

$$
\alpha(t) V^*(\mathbf{x}, t) = \max_{\mathbf{u}} \left[ R(\mathbf{x}, \mathbf{u}, t) + V_t^*(\mathbf{x}, t) + V_{\mathbf{x}}^*(\mathbf{x}, t) \, f(\mathbf{x}, \mathbf{u}, t) \right.
$$

$$
\left. + \frac{1}{2} \operatorname{tr} \left\{ V_{\mathbf{xx}}^*(\mathbf{x}, t) \, G(\mathbf{x}, \mathbf{u}, t) \, G(\mathbf{x}, \mathbf{u}, t)^T \right\} \right], \tag{7}
$$

where $\alpha(t)$ can be recognized to be the hazard rate corresponding to the survival function $S(t)$. We define the r.h.s. of the HJB equation in Eq. (7) without the maximization w.r.t. the action as

$$
Q(\mathbf{x}, \mathbf{u}, t) := R(\mathbf{x}, \mathbf{u}, t) + V_t^*(\mathbf{x}, t) + V_{\mathbf{x}}^*(\mathbf{x}, t) \, f(\mathbf{x}, \mathbf{u}, t)
$$

$$
+ \frac{1}{2} \operatorname{tr} \left\{ V_{\mathbf{xx}}^*(\mathbf{x}, t) \, G(\mathbf{x}, \mathbf{u}, t) \, G(\mathbf{x}, \mathbf{u}, t)^T \right\},
$$

so that the optimal policy is given by $\pi^*(\mathbf{x}, t) = \arg\max_{\mathbf{u}} Q(\mathbf{x}, \mathbf{u}, t)$. Later on, we will consider hyperbolic discounting, for which the hazard rate $\alpha(t)$ is given by Eq. (3), i.e., $\alpha(t) = \alpha_0 / (\beta_0 + t)$.

**Algorithm 1:** Computation of the optimal value function and policy for non-exp. discounting

---

**Result:** Optimal value function $V^\psi(\mathbf{x}, t)$, optimal policy $\pi^\psi(\mathbf{x}, t)$
**Input:** Parameters $\boldsymbol{\theta}$ of the discount function, system model, number of iterations $K$
Initialize parameters $\psi$ of neural network for modeling $V^\psi(\mathbf{x}, t)$;
**for** $k \leftarrow 0$ **to** $K - 1$ **do**

> Sample a batch of states and time points $(\hat{\mathbf{x}}, \hat{t})_{i=\{1,\dots,N\}}$;
> Push $(\hat{\mathbf{x}}, \hat{t})_{i=\{1,\dots,N\}}$ through the network to obtain $\hat{V}^\psi_{i=\{1,\dots,N\}}$;
> Use back-propagation to compute $\hat{V}^\psi_{\mathbf{x}i}, \hat{V}^\psi_{ti}, \hat{V}^\psi_{\mathbf{xx}i}$;
> Evaluate $E(\hat{V}^\psi_i, \hat{\mathbf{x}}_i, \hat{t}_i)$ in Eq. (8) and determine maximizing action $\hat{\mathbf{u}}_i$ for $i = 1, \dots, N$;
> Use back-propagation to compute the gradient of $\sum_i E(\hat{V}^\psi_i, \hat{\mathbf{x}}_i, \hat{t}_i)^2$ w.r.t. $\psi$ ;
> Update $\psi$ using the gradient ;

**end**
**return** $V^\psi(\mathbf{x}, t), \pi^\psi(\mathbf{x}, t) = \arg\max_{\mathbf{u}} Q(\mathbf{x}, \mathbf{u}, t)$

---

## 4.3 Solving the HJB equation

In order to solve the HJB equation in Eq. (7), which is a PDE, we apply a collocation-based method [28, 29, 31]. To do so, we first reformulate the HJB equation as

$$
E(V, \mathbf{x}, t) := -\alpha(t)V(\mathbf{x}, t) + \max_{\mathbf{u}} \Bigg[ R(\mathbf{x}, \mathbf{u}, t) + V_t(\mathbf{x}, t) + V_{\mathbf{x}}(\mathbf{x}, t) \, f(\mathbf{x}, \mathbf{u}, t)
$$
$$
+ \frac{1}{2} \operatorname{tr} \left\{ V_{\mathbf{xx}}(\mathbf{x}, t) \, G(\mathbf{x}, \mathbf{u}, t) \, G(\mathbf{x}, \mathbf{u}, t)^T \right\} \Bigg] = 0, \tag{8}
$$

and use a function approximator $V^\psi(\mathbf{x}, t)$ for $V^*(\mathbf{x}, t)$. The parameters $\psi$ of the approximator can be determined by sampling random states $\hat{\mathbf{x}}_i$ and time points $\hat{t}_i$ and minimizing $\sum_i E(V^\psi, \hat{\mathbf{x}}_i, \hat{t}_i)^2$ w.r.t. $\psi$. Calculating the derivatives $V^\psi_{\mathbf{x}}(\mathbf{x}, t), V^\psi_t(\mathbf{x}, t), V^\psi_{\mathbf{xx}}(\mathbf{x}, t)$ and differentiating the objective function w.r.t. $\psi$ is straightforward via automatic differentiation when choosing a neural network as function approximator [71]. As $t$ is not bounded, we need to choose a reparametrization of $t$ which maps all $t$ to the interval $[0, 1)$ before feeding the values into the network. More details about the implemented parametrization and application of the collocation method are provided in Appendix D. The complete algorithm to learn the value function and policy is provided in Algorithm 1.

## 4.4 Inverse reinforcement learning for inferring the discount function

When analyzing human behavior, one might be interested in learning the underlying discount function that led to a set of observed choices. In contrast to standard inverse reinforcement learning settings, where the goal is to learn the underlying reward function $R$, here we assume that the reward function is given and the discount function $S(t)$ is unknown and needs to be inferred. To learn the function, we assume a parametric form with parameters $\boldsymbol{\theta}$.

As given data, we assume the states and time points at which the subject switches from one action to another one, i.e., $\mathcal{D} = \{\mathbf{x}, \mathbf{u}^-, \mathbf{u}^+, t\}_{i=1\dots N}$, describing that in state $\mathbf{x}$ at time $t$ the action $u^-$ is switched to $u^+$. The observed decision maker is assumed to use the optimal policy $\pi(\mathbf{x}, t) = \arg\max_{\mathbf{u}} Q(\mathbf{x}, \mathbf{u}, t)$. Shortly before and after switching time $t$, we have $Q(\mathbf{x}(t-\Delta t), \mathbf{u}^-, t-\Delta t) > Q(\mathbf{x}(t-\Delta t), \mathbf{u}^+, t-\Delta t)$ and $Q(\mathbf{x}(t+\Delta t), \mathbf{u}^-, t+\Delta t) < Q(\mathbf{x}(t+\Delta t), \mathbf{u}^+, t+\Delta t)$, respectively, indicating that before $t$ action $\mathbf{u}^-$ is preferred over $\mathbf{u}^+$ and vice versa afterwards. By letting $\Delta t \to 0$, one finds $Q(\mathbf{x}(t), \mathbf{u}^-, t) = Q(\mathbf{x}(t), \mathbf{u}^+, t)$. A sensible objective for the inverse problem is therefore to minimize $\sum_i F(\mathbf{x}_i, \mathbf{u}_i^-, \mathbf{u}_i^+, t_i)$, defined as the squared difference of both Q values, i.e.,

$$
F(\mathbf{x}, \mathbf{u}^-, \mathbf{u}^+, t) = \left[ Q(\mathbf{x}, \mathbf{u}^-, t) - Q(\mathbf{x}, \mathbf{u}^+, t) \right]^2
$$
$$
= \Bigg[ R(\mathbf{x}, \mathbf{u}^+, t) - R(\mathbf{x}, \mathbf{u}^-, t) + V_{\mathbf{x}}(\mathbf{x}, t) \left( f(\mathbf{x}, \mathbf{u}^+, t) - f(\mathbf{x}, \mathbf{u}^-, t) \right) \tag{9}
$$
$$
+ \frac{1}{2} \operatorname{tr} \left\{ V_{\mathbf{xx}}(\mathbf{x}, t) \left( G(\mathbf{x}, \mathbf{u}^+, t) \, G(\mathbf{x}, \mathbf{u}^+, t)^T - G(\mathbf{x}, \mathbf{u}^-, t) \, G(\mathbf{x}, \mathbf{u}^-, t)^T \right) \right\} \Bigg]^2,
$$

**Algorithm 2:** Computation of the gradient of $F$ w.r.t. $\boldsymbol{\theta}$ for inferring the discount function

---

**Result:** Gradient of $F$ (Eq. (9)) w.r.t. $\boldsymbol{\theta}$ Eq. (10)
**Input:** Parameters $\boldsymbol{\theta}$ of the discount function, system model, number of iterations $K$
Determine $V^\psi(\mathbf{x}, \mathbf{u}, t)$ using Algorithm 1 ;
Initialize parameters $\phi$ of neural network for modeling $V_{\boldsymbol{\theta}}^\phi(\mathbf{x}, t)$;
**for** $k \leftarrow 0$ **to** $K-1$ **do**
    Sample a batch of states and time points $(\hat{\mathbf{x}}, \hat{t})_{i=\{1,\ldots,N\}}$;
    Push $(\hat{\mathbf{x}}, \hat{t})_{i=\{1,\ldots,N\}}$ through the networks to obtain $\hat{V}_{\boldsymbol{\theta} i=\{1,\ldots,N\}}^\phi$ and $\hat{V}_{i=\{1,\ldots,N\}}^\psi$;
    Use back-propagation to compute $(\hat{V}_{\boldsymbol{\theta}}^\psi)_{\mathbf{x}i}, (\hat{V}_{\boldsymbol{\theta}}^\psi)_{ti}, (\hat{V}_{\boldsymbol{\theta}}^\psi)_{\mathbf{xx}i}$;
    Evaluate $\sum_i H(\hat{V}_i^\psi, \hat{V}_{\boldsymbol{\theta} i}^\phi, \hat{\mathbf{x}}_i, \hat{t}_i)^2$ in Eq. (11) for $i = 1, \ldots, N$;
    Use back-propagation to compute the gradient of $\sum_i H(\hat{V}_i^\psi, \hat{V}_{\boldsymbol{\theta} i}^\phi, \hat{\mathbf{x}}_i, \hat{t}_i)^2$ w.r.t. $\phi$ ;
    Update $\phi$ using the gradient ;
**end**
**return** $\mathrm{d}F/\mathrm{d}\boldsymbol{\theta}$ by evaluating Eq. (10) using $V_{\boldsymbol{\theta}}^\phi(\mathbf{x}, t)$

---

w.r.t. the parameters $\boldsymbol{\theta}$ of the discount function. The optimization needs to take Eq. (8) as a constraint into account to ensure that the HJB equation is fulfilled. Note that Eq. (9) depends indirectly on the parameters $\boldsymbol{\theta}$ through the definition of the value function. However, the objective function $F$ could also directly depend on $\boldsymbol{\theta}$, we consider the general case in the following. In principle, the state at which the switching occurs also depends on $\boldsymbol{\theta}$. Nevertheless, we will assume that the parameters have only a minor influence on the states of switching, so that terms including the variation of $\mathbf{x}$ w.r.t. $\boldsymbol{\theta}$ can be neglected.

**Gradient computation**

For the minimization, it is useful to determine the gradient of the objective function $F$ w.r.t. $\boldsymbol{\theta}$, which can be formulated using the chain rule for total derivatives as

$$
\begin{aligned}
\frac{\mathrm{d}F}{\mathrm{d}\boldsymbol{\theta}} &= F_{\boldsymbol{\theta}} + F_V V_{\boldsymbol{\theta}} + F_{V_{\mathbf{x}}} (V_{\mathbf{x}})_{\boldsymbol{\theta}} + F_{V_{\mathbf{xx}}} (V_{\mathbf{xx}})_{\boldsymbol{\theta}} \\
&= F_{\boldsymbol{\theta}} + F_V V_{\boldsymbol{\theta}} + F_{V_{\mathbf{x}}} (V_{\boldsymbol{\theta}})_{\mathbf{x}} + F_{V_{\mathbf{xx}}} (V_{\boldsymbol{\theta}})_{\mathbf{xx}} .
\end{aligned}
\tag{10}
$$

By switching the order of the partial derivatives in the last step, we have obtained an expression for the gradient depending on $V_{\boldsymbol{\theta}}$. The partial derivative $V_{\boldsymbol{\theta}}$ measures the influence of the discounting parameters on the value function and is not straightforward to evaluate. We can get a PDE for this quantity by observing that the constraint $E(V^*, \mathbf{x}, t)$ in Eq. (8) is zero everywhere for the optimal value function $V^*$ and therefore its derivative also needs to be zero. This method is known as the forward sensitivity method [72] and provides an additional PDE for the desired gradient $V_{\boldsymbol{\theta}}$:

$$
\begin{aligned}
0 = \frac{\mathrm{d}E}{\mathrm{d}\boldsymbol{\theta}} &= E_{\boldsymbol{\theta}} + E_V V_{\boldsymbol{\theta}} + E_{V_t} (V_t)_{\boldsymbol{\theta}} + E_{V_{\mathbf{x}}} (V_{\mathbf{x}})_{\boldsymbol{\theta}} + E_{V_{\mathbf{xx}}} (V_{\mathbf{xx}})_{\boldsymbol{\theta}} \\
&= E_{\boldsymbol{\theta}} + E_V V_{\boldsymbol{\theta}} + E_{V_t} (V_{\boldsymbol{\theta}})_t + E_{V_{\mathbf{x}}} (V_{\boldsymbol{\theta}})_{\mathbf{x}} + E_{V_{\mathbf{xx}}} (V_{\boldsymbol{\theta}})_{\mathbf{xx}} =: H(V, V_{\boldsymbol{\theta}}, \mathbf{x}, t)
\end{aligned}
\tag{11}
$$

To solve the PDE and obtain $V_{\boldsymbol{\theta}}$, we can use the same procedure as in Section 4.3. To obtain the gradient, first one has to solve the HJB equation Eq. (7) to obtain $V^\psi$, then Eq. (11) to determine $V_{\boldsymbol{\theta}}^\phi$. Afterwards, Eq. (10) gives the gradient to update the parameters $\boldsymbol{\theta}$ of the discount function. The quantities $V_{\boldsymbol{\theta}}, (V_{\boldsymbol{\theta}})_{\mathbf{x}}, (V_{\boldsymbol{\theta}})_{\mathbf{xx}}$ for the derivative be computed via automatic differentiation as in Section 4.3. The complete algorithm for computing the gradient is presented in Algorithm 2.

## 5 Experiments

We tested our derived method on two simulated problems with a random termination time following a hyperbolic survival function. First, we solved the HJB equation in Eq. (7) using the collocation method (Algorithm 1) and computed the optimal policy. Then, for applying the inverse reinforcement learning method, we evaluated the objective function in Eq. (9) and computed gradients for different parameter sets $\boldsymbol{\theta} = [\alpha_0, \beta_0]$ of the hyperbolic discount function in Eq. (2) using Algorithm 2. For

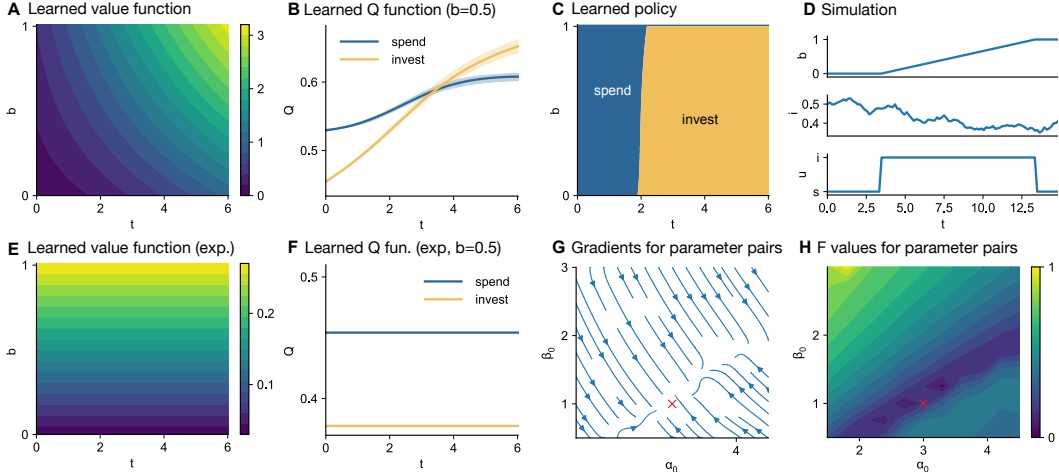

Figure 1: **Results for the investment problem. A** Learned value function for account balances and time points for hyperbolic discounting. **B** Learned Q function for both actions over time (median and quantiles for 10 runs). **C** Learned policy showing preference reversals for all account balances except for $b = 1$. **D** Simulation with account balance (top), interest rate (middle), and action (bottom) over time. **E** Learned value function for an exponential discount function, which is constant over time. **G** Learned Q function for an exponential discount function. **G** Gradients obtained for parameter pairs $[\alpha_0, \beta_0]$ of the discount function given simulated data on a $11 \times 11$ grid. The parameters used to generate the simulated trajectories are indicated by a red cross. **H** Values of the objective function $F$ (Eq. (9)) for parameter pairs on a $11 \times 11$ grid when applying the IRL method.

generating data, we randomly sampled starting states and determined the time points at which subjects would switch their action. Afterward, the determined time points were distorted by Gaussian noise.

All proposed methods were implemented in Python using the PyTorch framework [71] and are available online[1]. The used hyperparameters are listed in Appendix F. As tasks, we considered an investment problem and a problem controlling a point on a line. In the following we provide a brief overview of the considered problems, more details can be found in Appendix E.

## 5.1 Experimental tasks

In the investment problem, we model a subject having to decide whether to invest her or his income in the bank account leading to future interests (rewards) or to spend the money for immediate reward. We model the state as the current balance of the bank account as well as the current interest rate. When the money is spent (action *spend*), the subject receives rewards with a rate of 0.1 but the balance of the account remains unchanged. When the income is invested (action *invest*), the balance of the bank account increases with a rate of 0.1 but there is no additional reward. In both cases the subject receives rewards through interests, being proportional to the current balance on the bank account. We assume that the interest rate varies over time following a Gaussian diffusion model. To keep the state bounded, we model a maximum balance for the account.

In the line problem, the task is to control a point along a line. The state consists of the current position of the point. Possible actions are *left* and *right*, which move the point to the respective direction, as well as *stay*, which does not move the point at all. When moving the point, there is a Gaussian diffusion on the position and one has to pay a small action cost (negative reward). We model a state-dependent reward modeling a high reward for distant states on one side and a low reward for close states on the other side. This task can be seen as a continuous extension of the classical discounting tasks. A more detailed description of the functional form is provided in Appendix E.

---

[1] https://git.rwth-aachen.de/bcs/nonexp-rl

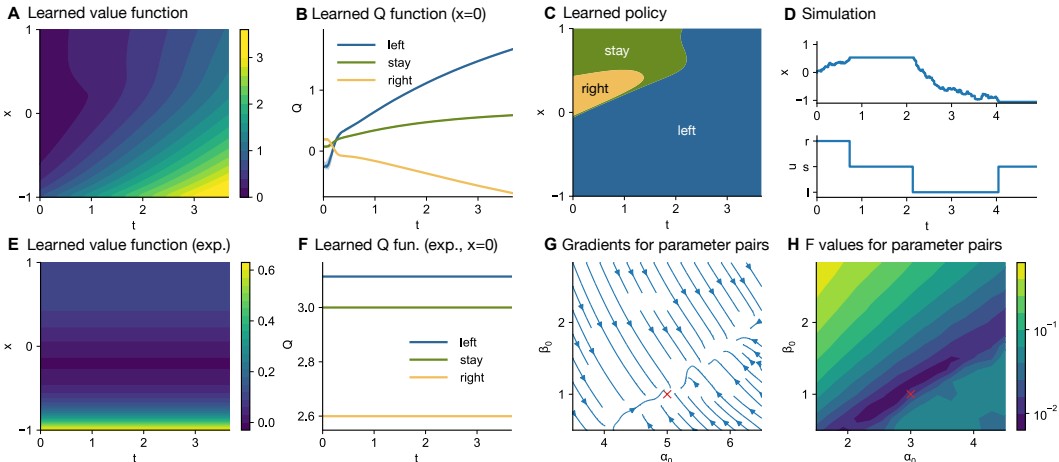

Figure 2: **Results for the line problem. A** Learned value function for hyperbolic discount function. **B** Learned Q function for each action over time (median and quantiles for 10 runs) **C** Learned policy showing preference reversals. **D** Simulation showing the state (top) and action (bottom) over time. **E** Learned value function for an exponential discount function. **G** Learned Q function for an exponential discount function. **G** Gradients obtained for parameter pairs $[\alpha_0, \beta_0]$ on a $15 \times 15$ grid given simulated data. The parameter used to generate the simulated trajectories is indicated by a red cross. **H** Values of the objective function of the IRL methods for parameter pairs on a $15 \times 15$ grid.

## 5.2   Results

The proposed method in Algorithm 1 produces plausible value functions for the considered problems. Figure 1 A shows the learned value function for the investment problem. The values are increasing with account balance (b), as expected rewards through interests become higher. Further in time, the value also increases, as the hazard rate can be assumed to be lower and one is expected to collect rewards for a longer time. Figure 1 B and C shows the learned Q function and policy, respectively. While it is advantageous to spend the income in the beginning, preference reversal occurs when the risk of termination is assumed to be relatively low, and investing becomes more attractive. The simulation in Fig. 1 D also reflects this behavior. For comparison, when assuming exponential discounting (Fig. 1 E and F), the value and Q function do not show preference reversal and remain constant over time. With regard to the inverse approach, Fig. 1 G shows the computed gradients of the IRL objective function $F$ in Eq. (9) for different parameter pairs. One can observe that the gradients mostly point in the direction of the parameters used for simulation, indicated by a red cross. Figure 1 H shows the evaluated IRL objective function $F$ for the parameter pairs, correctly displaying low values in the area close to the true parameters.

For the line problem, the computed value function is depicted in Fig. 2 A. Also in this task, the learned value increases over time and is high in areas close to highly rewarding states. Figure 2 B and C shows the learned Q function and policy respectively. In the beginning when the risk could be potentially high, if not close enough to the large reward, it is best to move right to collect the smaller sooner reward. With time when the risk of termination decreases, it becomes increasingly advantageous to move left to collect the larger later reward. In between, there is a time span, during which it is best to stay: The moving cost is too high for moving right and shortly turning afterward, but the risk is not low enough for being worth it to move left to the higher reward. The simulation in Fig. 2 D presents a sampled trajectory, showing that the subject first moves right until the maximum smaller sooner reward at $x = 0.5$ is reached and stays there for some time. After the risk has decreased over time, the simulated subject finally moves right to collect the larger later reward – her or his preferences have reversed. As for the investment problem, the value and Q function (Fig. 2 E and F) do not model any time-varying behavior. With regard to the IRL problem, the gradients and function values of $F$ of Eq. (9) in Fig. 2 G and H also lead closely to parameter values used for the simulations.

# 6 Conclusion

In this work, we have proposed a method for reinforcement learning with non-exponential discount functions. The approach can be used to solve decision-making problems with an arbitrary end-time distribution and to model human discounting behavior. First, we have shown the conditions for which the problem is well-defined when using a hyperbolic discount function. Then, we derived a HJB-type equation providing conditions for the optimal time-dependent value function. We presented how the obtained PDE can be approximately solved using a collocation method, leading to the optimal policy. Further, we introduced an approach for the inverse problem, in which the discount function needs to be inferred given behavioral data. The application of our methods on two simulated problems led to plausible solutions, opening the way for further applications such as the use in human experiments.

**Limitations and future work** In our proposed methods, we assume a finite action space. While this assumption applies to many behavioral experiments, it can be a limitation for the application to classical optimal control. To extend the method to continuous control, one has to determine how the maximization problem in the HJB equation is solved. For strictly convex action costs, there have been approaches proposed that efficiently solve the maximization in the HJB equation [31], under some conditions even in closed-form [29]. Further, for Lipschitz-continuous controls, it is possible to approximate the control dynamics to avoid solving the optimization problem [73]. Another requirement of our method is that the model needs to be known. Model-free approaches, such as TD-error learning for general discount functions could be explored, in line with work such as [16]. Further, when approaching problems with large state spaces, the proposed collocation method is likely to converge slowly. For these cases, adapted collocation methods [30] or advantage updating approaches [23] could be considered instead. To apply our method to more advanced problems in financial engineering, such as modeling stocks with discontinuous returns [74], one has to consider general jump-diffusion processes. While an extension to these models is straightforward from a theoretical point, we here left it out to keep the focus on the handling of the discount function. Regarding our inverse reinforcement learning approach, we have assumed the states and time points of the action switches to be given. While this assumption is reasonable for many human behavior experiments, it might be interesting to learn discount functions for given discretized trajectories instead. Also, incorporating an extended timing model of human decisions [75] instead of fixed-variance Gaussian diffusion would be an interesting extension.

In the future, we are planning to apply our proposed methods in human experiments to get new insights to human discounting behavior. Characterizing individual human subjects by analyzing their behavior comes with the risk to be used with negative social impact. This matter can be counteracted by collecting only anonymized data for the application of our method.

## Acknowledgments and Disclosure of Funding

We thank Bastian Alt for insightful discussions and for providing feedback on the first draft. This work was supported by the high-performance computer Lichtenberg at the NHR Centers NHR4CES at TU Darmstadt and the project "Whitebox" funded by the Priority Program LOEWE of the Hessian Ministry of Higher Education, Science, Research and Art.

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
