# OpenReview forum: "Reinforcement Learning with Non-Exponential Discounting"
_NeurIPS.cc/2022/Conference — NeurIPS 2022 Accept_

### Official Review · Reviewer_e6eZ · 2022-06-23

**Rating:** 6
**Confidence:** 4
**Soundness:** 4 excellent
**Presentation:** 4 excellent
**Contribution:** 3 good

**Summary:**

This paper generalizes the HJB equation in the infinite time-horizon setting to arbitrary discounting schemes. Based on this HJB equation, a collocation-based deep learning algorithm is presented for reinforcement learning under hyperbolic discounting. Moreover, the authors study the problem of inferring the discount function from data, and demonstrate a deep learning algorithm for accomplishing this task. Lastly, the authors demonstrate the performance of both their RL and discount-function-learning algorithms in two simple domains.

**Questions:**

Comments and Questions
======================

  **Line 104**: "... with the interpretation that the reward becomes
  unavailable with hazard rate $\alpha(t)$" -- Should it not be the case
  that *all future rewards* become unavailable at that hazard rate? As
  written, it sounds like reward at time $t$ can become unavailable, but
  reward at time $t + dt$ can still be available.

  **Equation after Line 127**: I believe the dimensions are incorrect
  here. The term on the LHS as well as the first two terms on the RHS
  are scalar-valued, but the final term on the RHS (I believe) has
  dimensions $m\times m$ -- the Hessian has dimensions $m\times m$, and
  since $G(\mathbf{x},\mathbf{u})\in\mathbf{R}^m$,
  $GG^\top(\mathbf{x},\mathbf{u})$ should also have dimensions $m\times
  m$.  I believe there should be a trace operator on that final term.
  Also, it says
  $R:\mathcal{X}\times\mathcal{U}\times\mathbf{R}_0^+\to\mathbf{R}$, but
  in the equations we have the term $R(\mathbf{x}, u)$ -- an argument is
  missing (and I believe $u$ should be $\mathbf{u}$). Is the reward
  function time-homogeneous?

  **Theorem 1**: It would be helpful to have a refresher on the meaning of
  $\alpha_0$ in the statement of the theorem. Also, why does $R$ need to
  be bounded below in order for the integral to diverge? I suspect what
  was meant was

  > If $\alpha_0\leq 1$, the value function diverges
  > regardless of the boundedness of $R$.

  Also, I don't think it's correct to say that "the value function
  diverges". Divergence is a property of a sequence, not a function. I
  believe "the value function is not well-defined", like in the first
  part of the theorem, is better.

  **Line 275**: I suspect continuous action spaces can be supported
  without much trouble using a similar technique as given in [2], which
  only requires the assumption that the control signal is Lipschitz
  continuous.


Minor issues / grammatical errors
=================================

  **Line 59**: "Solutions approaches that solve the HJB equation directly,
  include..." -- that comma shouldn't be there.

  **Line 60**: When talking about path integral approaches, I believe "A
  Generalized Path Integral Control Approach to Reinforcement Learning"
  [1] deserves a mention.

  *Line 63*: Personally, I find this paragraph to be redundant given the
  introduction.


References
==========

  [1] Theodorou, Evangelos, Jonas Buchli, and Stefan Schaal. "A
  generalized path integral control approach to reinforcement learning."
  The Journal of Machine Learning Research 11 (2010): 3137-3181.

 [2] Kim, Jeongho, Jaeuk Shin, and Insoon Yang. "Hamilton-Jacobi Deep
  Q-Learning for Deterministic Continuous-Time Systems with Lipschitz
  Continuous Controls." J. Mach. Learn. Res. 22 (2021): 206-1.


**Limitations:**

Limitations have been adequately addressed.

**Strengths And Weaknesses:**

# Strengths
It was immediately evident that the paper is written in great English, making the text very easy and comfortable to follow for the most part. The authors should be commended for this.

I appreciate the authors' ambition to study this problem in continuous time, which is often overlooked. The collocation algorithm, while fairly standard, is not particularly common in RL, and I believe the RL community should be interested by it. Especially given the time-inhomogeneity of the value function with arbitrary discounting, the collocation method seems quite elegant.

Moreover, I do believe the "inverse RL" problem studied in this paper -- that being the problem of inferring the discounting function -- is an interesting one. I can imagine the algorithm presented in this paper to be impactful in economics / neuroscience / psychology.

# Weaknesses
There are a few mathematical issues and concerns, however they're all minor and surely they can be easily addressed.

I found that the discussion of experimental results is highly qualitative/speculative. Essentially, I don't find that the experimental results do a good job of convincing the reader that arbitrary discounting is valuable. It would have been nice, for example, to see an example where perhaps a nonstandard discounting scheme leads to a value function that is much more intuitively correct than that with an exponential discounting scheme.

Finally, I believe the novelty of the contributions is relatively limited. I do believe some of the contributions are valuable -- particularly the general discounting HJB equation -- however I believe the paper could have a lot more impact with a more developed experimental analysis.

---

> ### Author Response · Authors · 2022-08-01
> **Reply to reviewer e6eZ**
>
> In our experiments, we have compared the policies obtained by our method to the ones when assuming exponential discounting with a fixed hazard rate. In case of uncertainty about the hazard rate, it should be intuitive that preferences may change, depending on the current belief about the risk. In the investment example, when one is uncertain about the risk, it is clearly beneficial to first spend the income, but at a later point in time, when one can assume the risk to be low, it is advantageous to start investing. Also in the line problem, we observe the subject to become more farsighted with time and therefore displaying the type of preference reversal which has been extensively found in human experiments (see, e.g., ref. [3, 4, 5, 33, 34] in the manuscript). When applying an exponential discounting formalism (Fig. 1 E, F, Fig. 2 E, F in the manuscript), we can observe that it fails to capture these time-dependent preferences. We still agree that the rationality of preference reversals may seem not very intuitive at first sight. To improve clarity, we will extend the discussion of the results in section 5.2 with a more detailed explanation of this matter.
>
> In section 3.2, containing line 104, which you referenced, we give an introduction to the classical discounting formulation. In this setting, one considers only one reward, which is provided in the future and discounted over time. In the optimal control formulation, on the other hand, one assumes a state-dependent reward rate per time step, so multiple rewards can be collected. We understand that the sentence "We consider the setting in which a subject collects rewards" (line 100) may cause confusion on this. Therefore, we will change it to "[...] the setting in which a subject collects a single reward". Additionally, we will add some remarks in section 4 on how the setting there contrasts with section 3.2.
>
> Thank you for spotting the missing trace operator in equation 127. In further derivations and the derived generalized HJB, the operator is there, so this omission had no further consequences.
>
> Also, thank you for spotting the inconsistency between the standard optimal control formulation (section 3.3.) and the formulation in section 4.4. In standard optimal control (section 3.3), the reward function must not be time-dependent, as otherwise the value function would also depend on time. In our formulation in section 4, the reward function may depend on time as the value function needs to be time-dependent anyways. We have actually noted it this way in section 4 but unfortunately not commented on this fact. The same applies to drift and dispersion. We will add a comment on this to the beginning of section 4.
>
> Thank you for your recommendation regarding the refresher on $\alpha_0$. We will add it to the statement of the theorem.
>
> Concerning your question about whether $R$ needs to be bounded below for the integral to diverge: If $R$ was not bounded below and the reward zero at a certain state, the integral could be finite as well. Therefore, to be certain that the integral diverges, one needs an additional requirement (a softer one would also be possible, yet more complicated.)
>
> We are sorry for the incorrectly formulated expression "the value function diverges". We mistakenly thought about the expression "the integral diverges", which led to this mistake. We will write "not well-defined" instead and add a short explanation that the integral diverges to the end of the proof.
>
> Thank you also for the comment regarding continuous action spaces. Supporting them should indeed be possible following the approach you suggested. We have already mentioned the possibility of extension for the case of strictly convex action costs. In this case, one can maximize the HJB equation efficiently (following references 27 and 29 in the manuscript) and keep the collocation method. We will add your suggestion to the future work part of the conclusion.
>
> Lastly, thank you for pointing out the minor issues. We will correct them in the final version of the manuscript and cite the publication suggested by you when mentioning path integral approaches.

---

### Official Review · Reviewer_7uF8 · 2022-07-10

**Rating:** 7
**Confidence:** 4
**Soundness:** 4 excellent
**Presentation:** 3 good
**Contribution:** 3 good

**Summary:**

This paper introduces a reinforcement learning method for computing optimal policies for non-exponential discount functions in continuous-time environments, as well as an inverse reinforcement learning method for estimating the parameters of the discount function based on the learnt policy, assuming that the reward function itself is known. The approach relies on solving a HJB equation. The hyperbolic discount function is given particular attention, but the proposed methods apply to essentially any discount function.

Any discount function besides exponential discounting can induce preferences that are inconsistent over time. This leads to some ambiguity as to what it should mean for a policy or action to be "optimal" under such a discount function. In the literature on dynamic choice, there are three natural candidates which are often suggested: naive choice, sophisticated choice, and resolute choice. The notion of optimality that this paper relies on is analogous to resolute choice.

**Questions:**

The main questions that this paper raises for me are,

1. Can the proposed methods be used in discrete-time settings? I imagine the answer is yes, but I would have liked to see some concrete mention of how to do this (even if just a short remark).

2. How scaleable are the proposed methods? The experiments provided in the paper both involve very small environments, and the paper does not provide the time it took for the algorithms to converge in these environments. The authors do remark that "when approaching problems with large state spaces, the proposed collocation method is likely to converge slowly", but I found that I did not get a good sense of what exactly this means in practice. Do these methods only work in toy environments, with states that have one or two parameters, or would they be applicable to environments whose states have at least a few thousand parameters or so?

3. Computing a (resolute) optimal policy, for a given discount function, is equivalent to computing an optimal policy in a modified MDP that, at each time step, terminates with a certain probability, and where the state-space includes the time (this is also pointed out by this paper in Appendix H). Moreover, if we include the time in the state space, then we could alternatively also just directly modify the reward function to reflect the discounting. Does the reinforcement learning method proposed in this paper provide any benefits over either of these two approaches, and if so, what are they? It seems to me that either of these two approaches would be easier to implement, and it would moreover be straightforward to combine either of them with arbitrary state-of-the art methods. I would certainly like to see some discussion of if and why Algorithm 1 improves on these two strategies. If it does not, then that seems to reduce the impact of the paper (although the paper of course also includes an IRL method, which I imagine was the original motivation for this work, given the final remark of the paper).

4. There has been some work on inverse reinforcement learning for non-exponential discounting (see "Irrationality can help reward inference" by Chan et al, and "Learning the Preferences of Ignorant, Inconsistent Agents" by Evans et al). Moreover, in the fields of preference elicitation and behavioural economics, there has been a very large amount of work on inferring discount rates from behaviour. How does the inverse reinforcement learning method proposed in this paper relate to and improve upon what is provided by this earlier work? In my opinion, an extensive discussion of this question has to be included in the paper.

If these points are addressed (especially 3 and 4), then that could convince me to increase my rating. I would be particularly keen to see a more detailed comparison between the proposed IRL algorithm and existing methods from the preference elicitation literature.

**Limitations:**

I find that this paper provides a fully satisfactory discussion of its limitations, as well as of its potential negative societal impacts.

**Strengths And Weaknesses:**

Originality:

There has been some work on reinforcement learning with non-exponential discounting, but that work is fairly preliminary, and this paper certainly approaches the problem from a different angle than those earlier papers. However, I have some questions about the benefits of the approach proposed in this paper, see my question 3 below. On the inverse reinforcement learning side, there has also been some earlier work concerning non-exponential discounting, but to my knowledge, that work only concerns the problem of inferring the reward function when the discount function is known, whereas this paper concerns the problem of inferring the discount function when the reward function is known. The inverse reinforcement learning method is therefore certainly novel in that context. However, in the economics literature, there has been quite a lot of work on the problem of inferring a discount function based on consumer behaviour. I do not know how the inverse reinforcement learning method proposed in this paper relates to that earlier work in economics, see my question 4 below.

Quality:

The work presented in this paper is of high quality.

Clarity:

I found this paper to be very clear, except for my questions 1-2 below.

Significance:

My guess is that the problem which this paper addresses probably would be perceived as somewhat niche by the broader machine learning research community. However, it is very relevant to fields such as e.g. preference elicitation. My personal opinion is therefore that the problem is significant.

---

> ### Author Response · Authors · 2022-08-01
> **Reply to reviewer 7uF8 (Part 1)**
>
> The proposed HJB equation is not immediately applicable to discrete time, but the derivation of a discrete-time Bellman Equation is analogous to the one in the manuscript. For that, one needs to adapt the definitions slightly, and importantly, all rates become conditional probabilities (e.g., the hazard rate becomes a hazard probability). The resulting general-discounting Bellman equation compares to the standard Bellman equation as our derived HJB equation to the standard one (e.g., ref. [14] in the manuscript). In particular, as in the continuous-time case, the Bellman equation becomes time-dependent and depends directly on the conditional hazard probability function. For solving the Bellman equation, one still needs a continuous time representation for the function approximator. As there are infinitely many possible time values, it is not possible to use a discrete representation in the implementation (e.g., a table), which limits the advantages of the discrete-time formulation. We will add a derivation of the discrete-time Bellman equation and comments on the solution procedure to the appendix of the final version of the manuscript.
>
> We have listed the training durations in Appendix F of the manuscript. For the investment problem, training took roughly 50 minutes while for the line problem it took 30 min on an Intel Xeon Platinum 9242 Processor with 8 cores. We have not optimized the hyperparameters (network size, learning rate, etc.) but checked for convergence. We would expect the proposed method to fail for high-dimensional problems (e.g., a few thousand dimensions as you have mentioned) as it is a collocation-based method and, as mentioned in the limitations section, leave the development of scalable methods for future work. Nevertheless, the method in its current form should apply to most experimental settings in psychology and economics, in which usually only low-dimensional variables (such as income, reward magnitude, taxes) are considered.
>
> Thank you for proposing the two alternative strategies to solve the problem. One option to realize the first strategy would be to add a terminal state and modify the transition function accordingly. However, extending the continuous state space by a discrete termination state is not straightforward and would yield a hierarchical model, needing more advanced solution methods. Another option would be to consider the termination probability directly in the derivation of the HJB equation, i.e., to set all future reward with a certain probability to zero. For this, let us consider the formula between lines 155 and 156 in the manuscript. The immediate reward would be received with probability one, so the first term in the expectation would eventually end up as the same one for $\Delta t \to 0$. The second term would need to be multiplied with the probability of not terminating, which is the probability of surviving provided that one survived until this time point. This is actually the same factor as in our derivation and the resulting HJB equation would be equal.
> The second approach you have proposed would extend the state by the time and pull the discount function into the reward function. This case would be equivalent to our definition of the value function without $S(t)$ in the denominator. The disadvantage of this formulation is that for $t$ going to infinity, the value function approaches zero, introducing numerical problems and making it hard to converge (we have actually tried out this formulation at the beginning of the development of our method). Moreover, this method also does not provide any advantages from a computational viewpoint as one still needs to solve the time-dependent HJB equation with the augmented state.
> Finally, as you have mentioned, the original motivation was indeed to do IRL, which would still need special treatment, as the usually-applied maximum entropy formulations would apply there.

---

> ### Author Response · Authors · 2022-08-01
> **Reply to reviewer 7uF8 (Part 2)**
>
> We are sorry for not having included a more detailed discussion of related work in the fields of preference elicitation and behavioral economics. In the paper by Chan et al. that you have suggested, the authors make a rather crude approximation of the value function for the hyperbolic discounting case, which was originally proposed in the work of Alexander & Brown (ref. [15] in the manuscript). As in a hyperbolic discounting setting, the value tends to increase over time (as the expected hazard rate decreases), they replace the time with the value function and obtain a value function that becomes independent of time. While this approach can be used to generate behavior that looks somehow similar to one of a hyperbolically-discounting subject, one cannot consider a specific discount function. Most importantly, they do not approach the problem of inferring the discount function (which would anyways not be possible using this approximation) but aim at inferring the reward function.
> The latter also applies to the paper by Evans et al. Further, in that work, the authors consider the discrete-time and discrete-state setting with a finite horizon, in contrast to our work.
> We address your general question about related work in the fields of preference elicitation and behavioral economics in our comment to all reviewers above.

---

### Official Review · Reviewer_snXr · 2022-07-13

**Rating:** 5
**Confidence:** 2
**Soundness:** 3 good
**Presentation:** 3 good
**Contribution:** 3 good

**Summary:**

This paper provides a mathematical derivation of the HJB equation for a general discount factor (i.e., discount factor that is a function of time, as opposed to the traditionally fixed exponential discount rate) in the continuous-time optimal control / reinforcement learning setting. The paper provides two algorithms: one for solving the HJB (computing the value function), and the other for inverse determination of the discount factor parameters given a known reward and time of preference reversal. The validity of the derivations is demonstrated in two toy experiments.

**Questions:**

See main review.

**Limitations:**

Yes, I think the limitations are addressed (but see main review).

**Strengths And Weaknesses:**

**Significance/novelty:** Despite the authors' positioning of this as a "reinforcement learning" paper, it appears to be more of an optimal control / behavioral economics paper. The experiments do not use any baselines, and merely exist as empirical validations of the derived HJB + algorithms. The main practical application that is claimed is for the study of human discounting, rather than as work that has applications to training RL agents. This is fine, but it is a bit outside my expertise, and perhaps a bit incongruent with the framing RL framing (e.g., Alg 1. does not take in data, which I would consider to be the defining feature of RL, but rather the system model), and the data used for the "Inverse RL" task is rather crude. I would consider the IRL bit to be more of an exercise in elicitation.

In that regard, I feel the connections to related work are underdeveloped, and I'm not sure how significant the problem treated is. I'm left with the following questions:

- What is the closest work in Optimal Control, and what are the specific contributions of this beyond that? How standard vs non-obvious is the derivation here really? I haven't worked with PDEs/HJB before, so I'm not sure if the derivation is basic, and I would not be surprised if this exists, at least, in a very close form. E.g., In Zou et al., Finite horizon consumption and portfolio decisions with stochastic hyperbolic discounting, 2014 [Not cited], "*Section 3 derives the HJB equation for sophisticated individuals with stochastic hyperbolic discounting.*", and a Scholar search for *"hyperbolic discounting" HJB* reveals 266 results. I would want to understand how this contribution fits into this other work, rather than the high level, no-contrast description provided at lines 57-68.

- Is there really no comparable elicitation method from behavioral economics that could be contrasted against? A Google Scholar search for *"hyperbolic discounting" elicitation* reveals over 5000 results. Several on the first page appear very relevant and aren't cited. Certainly if economists have observed that humans follow a hyperbolic discounting schedule, they must have ways of eliciting this (i.e., doing something comparable to the Inverse RL algorithm proposed in Subsection 4.4) --- how is this problem treated in behavioral economics, and how does this work compare?

- Is this work relevant at all to current (deep) reinforcement learning research? For example, I would say the discussion in Fedus et al. (2019) is complete / well positions the usage of hyperbolic discounting for purposes of reinforcement learning. The Pathworld experiments and the discussion surrounding them make clear when hyperbolic discounting might be useful. It also shows a practical application of the techniques to a relevant modern baseline (albeit the effectiveness has nothing to do with hyperbolic discounting).

**Quality/Clarity** I did not verify the derivations/proofs, and am not familiar with the algorithm techniques (collocation method) to comment on the technical validity. That said, the paper reads well. At a high level (I did understand all the notation, and have reviewed a bit on HJB), the math checks out / "looks correct". The writing is overall solid. The introduction could be more concise, with more focus on the relevance of the present work rather than discounting / hyperbolic discounting as a whole. And see my comments on related work above.

**Bottom line**: While this looks correct and is well written / clear, I am a bit wary of recommending accept given the positioning and lack of clear contrast against past work. I would be willing to change my view given confidence from other reviewers on the novelty/positioning issues, and a solid rebuttal from the authors.

EDIT: I found the authors rebuttal responsive with regards to the questions above and increased my score accordingly.

---

> ### Author Response · Authors · 2022-08-01
> **Reply to reviewer snXr**
>
> We agree that our paper is very close to the fields of optimal control and behavioral economics. By positioning our work in the field of reinforcement learning, we have followed a widely-adopted terminology, which is also given in the book "Reinforcement learning and optimal control" (Bertsekas, 2019), defining reinforcement learning as "solution methods that rely on approximations to produce suboptimal policies with adequate performance". As the policy cannot be directly determined but is learned via samples (see collocation method), we believe that the naming is appropriate. Nevertheless, we will add the term "model-based" to the abstract and introduction to avoid any confusion.
>
> Regarding the lack of baselines in the experiments, we would like to highlight that we actually have used the exponentially-discounted HJB equations as a baseline for comparison (Fig 1 E, F and Fig 2 E, F). And indeed, as you ask, to the best of our knowledge, there are no other methods for solving the hyperbolically-discounted problem with sequential actions yielding a time-varying policy.
>
> Regarding your question on related work in optimal control, the HJB equation has been commonly studied in the exponentially-discounted case (see e.g., ref. [14] and [56] in the manuscript). Other discount functions have received comparably little attention and if so, only special cases have been regarded. The closest works in this field consider quasi-hyperbolic discounting (e.g. ref. [42, 43, 44] in the manuscript), which also the work you suggested belongs to. Quasi-hyperbolic discounting assumes that the total time period can be divided into two subintervals: In the first, the subject discounts exponentially with a fixed rate, whereas in the second, one is assumed to discount exponentially with the same rate and further discounts with an additional constant factor. As the expected total discounted reward can be decomposed into two (almost) exponentially-discounted parts, the final HJB equation does not need to depend on a generally time-dependent hazard rate. When solving the HJB equation, these approaches then consider a policy that remains constant over time (such as ref. [16] of the manuscript).
> In our work, we consider a more general setting in which we are not limited to the exponential or quasi-hyperbolic form. Our derived formulation of the HJB equation depends on the time-dependent hazard function, corresponding to a general discount function, and we obtain a time-dependent policy for the infinite horizon case. We would therefore not label our result as "standard" or "obvious".
> We are sorry that the discussion of this matter fell rather short in the manuscript, so we will extend the introduction and the related work with a more detailed discussion in the final version. In addition to the work suggested by you, we will also cite and discuss the following works:
>
> Chunxiang, A., Zhongfei Li, and Fan Wang. "Optimal investment strategy under time-inconsistent preferences and high-water mark contract." Operations Research Letters 44.2 (2016): 212-218.
>
> Chen, Shumin, Zhongfei Li, and Yan Zeng. "Optimal dividend strategy for a general diffusion process with time-inconsistent preferences and ruin penalty." SIAM Journal on Financial Mathematics 9.1 (2018): 274-314.
>
> Grenadier, Steven R., and Neng Wang. "Investment under uncertainty and time-inconsistent preferences." Journal of Financial Economics 84.1 (2007): 2-39.
>
> Regarding your question on related work in the field of behavioral economics, please see the comment to all reviewers.
>
> On your question about the relevance to "current (deep) RL" research, we think that the use of non-exponential discount functions should indeed be limited to applications in which there is a good reason to do so. For applications like Atari games or robot control, we believe that there is no significant advantage of using a hyperbolic discount function to model time preference and it makes the problem more complicated (this is also in line with the reviews of the paper by Fedus et al.). However, if one needs to consider a certain end-time distribution in an environment as in applications to psychology, economics, and finance, we believe that our theory and method can be of great benefit to the community. Additionally, the derived IRL method is based on the here developed theory.

---

> > ### Comment · Reviewer_snXr · 2022-08-08
> > **Response**
> >
> > Thank you, I found your comments are responsive and have increased my score to 5 (I am not comfortable raising it higher at the moment due to (a) limited understanding on my part, and (b) lack of revision currently, although this paper could very well be in 6-7 range).

---

### Official Review · Reviewer_MBPJ · 2022-07-14

**Rating:** 5
**Confidence:** 3
**Soundness:** 2 fair
**Presentation:** 3 good
**Contribution:** 3 good

**Summary:**

The paper studies continuous-time reinforcement learning with general discounting rewards and hyperbolic survival functions under the framework of optimal control (i.e., a generalization of the HJB equation). The extension of the HJB equation leads to a value-based learning method, and the authors discussed how to apply inverse reinforcement learning to infer the discount function. The authors conduct experiments on two simulated problems and demonstrate the effectiveness of their approaches.

**Questions:**

See the weakness part.

**Limitations:**

yes.

**Strengths And Weaknesses:**

Advantages:

- The technique contribution of the paper is good. Generalization of the HJB equation makes sense to me and studying continuous time RL is also interesting.
- The motivation of inverse RL is very strong to me, and I think the proposed method that exploits the structure of the dynamic model is novel.

Weakness:

I’d say that the paper seems over-claimed. The title suggests a general reinforcement learning algorithm, and the paper claims arbitrary discount functions. Unfortunately, these arguments do not hold in my view.

Regarding reinforcement learning,  I would say that the paper only studies a very limited set of RL problem. The requirement of knowing the forward model $f$ and $G$ makes the proposed approach more like a model-based RL or optimal control instead of reinforcement learning. Besides, I believe the method at least requires the state space to be continuous and the drift function twice differentiable so that the value function could be differentiable w.r.t. states. The latter assumption is not explicitly stated in the paper. And with the paper’s general discounting function, it is not easy to ensure that the value function, which is the solution to an ODE, is twice-differentiable w.r.t. the states (as required by the HJB). Additional assumptions like some lipschitz conditions (as in Picard–Lindelöf) might be necessary. Though the conclusion may hold under some mild conditions, I urge the authors to carefully examine those conditions for strictness.

Similarly, I appreciate the study of hyperbolic discounting, but I don’t think using it to claim an arbitrary distribution is proper. For general discounting, a bounded $R$ may not guarantee a valid value or value gradients. For example, considering a chaotic system, a slight change of the initial state would result in a huge difference in the future. When the discounting function is not carefully specified, the value may have an unbounded gradient in the end.

A minor issue is the lack of human experiments. The key motivation is to study human’s behavior. However, there are no human experiments in the paper. I think this will largely downgrade the contribution of the paper.

---

> ### Author Response · Authors · 2022-08-01
> **Reply to reviewer MBPJ**
>
> We agree that our proposed theory and method do not allow for any conceivable function to be used as discount function, as there are cases in which no solution exists.
> First, as mentioned in section 4.1, the objective of the discounted long-term reward is not well-defined for discount functions which make the integral in equation 6 of the manuscript diverge. For the hyperbolic discount function, we have derived the conditions in section 4.1. For other discount functions, given a bounded reward function, a sufficient condition is that the integral of the survival function needs to be bounded for the discounted long-term reward to be well-defined.
> A further requirement is that the stochastic process defined in section 3.3 has a strongly unique solution, which can be determined using the stochastic Picard iteration. For a strongly unique solution to exist, $f$ and $G$ need to grow at most linearly in $X$ and be Lipschitz continuous in the same variable. More details about this condition can be found for example in the book "Applied stochastic differential equations" (Särkkä & Solin, 2019).
> Then, we would like to clarify that the optimal value function is the solution of a PDE, not an ODE. The optimal value function is only in rare cases smooth enough to be a solution in the "classical" sense. Instead, one needs to regard a so-called "viscosity solution", which satisfies the HJB equation in an appropriate generalized sense, i.e., it needs to be a super- and sub solution. With this formulation, the optimal value function does not need to be everywhere differentiable but only locally Lipschitz. The conditions for the existence and uniqueness of a viscosity solution are discussed in the book "Controlled Markov Processes and Viscosity Solutions" (Fleming & Soner, 2006). A sufficient condition is that $f$ and $G$ are continuous with bounded continuous first derivatives w.r.t. $X$ and $t$, and $G$ bounded. $R$ and $S$ need to be continuous and grow at most polynomially in $X$ and $t$ in an absolute sense. Note that these requirements are fulfilled for the hyperbolic discount function and most discount functions of interest.
> Thank you for pointing out that we have not provided the exact technical requirements in the manuscript. We will add a more detailed discussion on these and give sufficient conditions for the general discount function in section 4.1.
>
> Reinforcement learning has become a wide field with a broad collection of different problem settings and solution algorithms. In general, RL is defined to be algorithms for solving the optimal control problem through some form of learning (Sutton & Barto, 2019).
> You are right that we consider the setting of model-based RL. While we think that a general discounting framework for model-free settings would be also an interesting direction, the theory for the considered setting is already quite complex. In nearly all discounting studies, subjects are given knowledge about the environment (commonly delays and rewards), which is why we think that this setting needs to receive the most attention.
> We agree that we can improve the abstract and introduction to clarify which reinforcement learning setting we consider exactly, so we will add this information there to avoid any confusion.
>
> Regarding the lack of human experiments, we share your opinion that for work with applications to economics and psychology, human experiments are generally advantageous. Considering the presented work, our main contribution is the derivation of the theory and an algorithm for solving the forward and inverse (model-based) reinforcement learning problem as well as validation through simulation.
> We think that the application of our method to human data would have a limited benefit in the scope of this paper to validate our method, as there is no ground truth to compare the inferred discount functions to. Still, for new insights regarding explanations of human behavior, human experiments are indispensable but require more elaborate experimental hypotheses.

---

> > ### Comment · Reviewer_MBPJ · 2022-08-07
> > **response to authors**
> >
> > Thank you for your response. I appreciate your clarification about technical requirements. Such kinds of conditions should be sufficient. I can raise my score to 5. However, I still think that either human experiments or experiments in higher dimension experiments are needed to make the paper strong in neurips submission.

---

### Author Response · Authors · 2022-08-01
**General comment to all reviewers**

First, we would like to thank all reviewers for their time spent with our manuscript and their helpful comments. We particularly appreciate the clarification questions that have helped sharpening our exposition and the detailed feedback about the computational methods. We would like to thank the reviewers also for judging our work to be of "high quality", "quite elegant", as well as "interesting", "impactful", and "certainly novel in that context".

Before providing one-to-one detailed answers to individual questions, we would like to clarify the scope and context of our contribution, particularly as two reviewers had questions about related work in the fields of behavioral economics and preference elicitation.
We are sorry for not having included a more detailed discussion of previous work on eliciting discount functions from behavior. Some relevant prior work in which discount functions have been estimated, are mentioned in the related work section (see e.g., references [33, 34, 36, 37] in the manuscript). But, in all these works known to us, only non-sequential decision-making tasks have been considered, i.e., for each independent trial only one decision has to be made. To make this point very clear: these problems address independent single decisions whether to select for example $x$ now or $y$ after a time $t$.
As a result, the planning problem becomes a very simple one (see first paragraph of section 3.2). It is then straightforward to get a non-parametric estimate of the discount function (e.g., Green & Myerson, 1996) or to determine the discount parameters via maximum likelihood (e.g., Schweighofer et al., 2006).
Exceptions to independent trials have been considered, e.g., in the work by Schweighofer et al. or Seinstra et al. (referenced below). In this line of work, decisions actually had an influence on the future (i.e., the remaining experiment time), but could be treated independently for data analysis when regarding the reward rate.

Instead, we consider the case in which subjects need to plan their decisions in a varying environment to maximize their total reward. To the best of our knowledge, there have not been any behavioral experiments conducted in this setting due to the lack of methods for data analysis. We therefore believe that our method will be of great interest for the development of computational models of human behavior, to analyze the discounting phenomenon in more naturalistic settings. In subsequent human experiments, we think it will be important to not only estimate discount functions given behavior (which we have shown via simulations in the current work) but also to validate hypotheses on the rationality of human discounting. As this matter is fairly intricate as well, we leave the conduction of human experiments as future work.

We will try to give a clearer overview of this paper's contributions in the introduction of the final version of the manuscript. Further, we will extend the related work section with sharper contrast to publications dedicated to methods for eliciting discount functions, especially in the fields of behavioral economics. To this end, we will additionally reference and discuss the following works:

Noor, Jawwad. "Hyperbolic discounting and the standard model: Eliciting discount functions." Journal of Economic Theory 144.5 (2009): 2077-2083.

Andersen, Steffen, et al. "Eliciting risk and time preferences." Econometrica 76.3 (2008): 583-618.

Coller, Maribeth, and Melonie B. Williams. "Eliciting individual discount rates." Experimental Economics 2.2 (1999): 107-127.

Richards, Jerry B., et al. "Determination of discount functions in rats with an adjusting‐amount procedure." Journal of the experimental analysis of behavior 67.3 (1997): 353-366.

Green, Leonard, and Joel Myerson. "Exponential versus hyperbolic discounting of delayed outcomes: Risk and waiting time." American Zoologist 36.4 (1996): 496-505.

Schweighofer, Nicolas, et al. "Humans can adopt optimal discounting strategy under real-time constraints." PLoS computational biology 2.11 (2006): e152.

Seinstra, Maayke Suzanne, Manuela Sellitto, and Tobias Kalenscher. "Rate maximization and hyperbolic discounting in human experiential intertemporal decision making." Behavioral Ecology 29.1 (2018): 193-203.

---

### Meta-Review · Area_Chair_kzLa · 2022-08-23

**Recommendation:** Accept
**Confidence:** Less certain

**Metareview:**

This paper studies various forms of discounting in continuous time, with a deep learning solution. One of the motivations for doing so is to broaden the range of inverse RL algorithms.

Overall, the reviewers appreciated the perspective taken in this paper and the proposed application of the idea in an IRL context. There was some discussion on whether actual human data experiments were necessary; I note that the experimental results are currently fairly preliminary. The use of the term "reinforcement learning" is also somewhat misleading given that this is closer to more traditional OR work, including the assumption that the MDP parameters are known. However, there was general agreement that this paper plays a useful role in bridging different fields and makes a good contribution.

The authors are encouraged to give a more complete discussion of how this work relates to other techniques such as preference elicitation.

**Award:**

No

---

### Decision · Program_Chairs · 2022-09-14

Accept